# Discovery of *Lebambromyia* in Myanmar Cretaceous Amber: Phylogenetic and Biogeographic Implications (Insecta, Diptera, Phoroidea)

**DOI:** 10.3390/insects12040354

**Published:** 2021-04-16

**Authors:** Davide Badano, Qingqing Zhang, Michela Fratini, Laura Maugeri, Inna Bukreeva, Elena Longo, Fabian Wilde, David K. Yeates, Pierfilippo Cerretti

**Affiliations:** 1Department of Biology and Biotechnologies “Charles Darwin”, Sapienza University of Rome, Piazzale A. Moro 5, 00185 Rome, Italy; 2Museum of Zoology, Sapienza University of Rome, Piazzale Valerio Massimo 6, 00162 Rome, Italy; 3State Key Laboratory of Palaeobiology and Stratigraphy, Nanjing Institute of Geology and Palaeontology and Center for Excellence in Life and Paleoenvironment, Chinese Academy of Sciences, 39 East Beijing Road, Nanjing 210008, China; qqzhang@nigpas.ac.cn; 4Institute of Geosciences, University of Bonn, 53115 Bonn, Germany; 5CNR-Nanotec (Rome Unit) c/o Department of Physics, Sapienza University of Rome, Piazzale A. Moro, 5, 00185 Rome, Italy; michela.fratini@gmail.com (M.F.); maugeri83@gmail.com (L.M.); innabukreeva@yahoo.it (I.B.); 6P. N. Lebedev Physical Institute, Russian Academy of Sciences, Leninskii pr., 119991 Moscow, Russia; 7Helmholtz-Zentrum Geesthacht, Institute of Materials Physics, Max-Planck-Strasse 1, 21502 Geesthacht, Germany; elena.longo@hzg.de (E.L.); fabian.wilde@hzg.de (F.W.); 8Australian National Insect Collection, CSIRO National Facilities and Collections, Black Mountain, Clunies Ross Street, Acton, Canberra, ACT 2601, Australia; david.yeates@csiro.au

**Keywords:** Eremoneura, Mesozoic, systematics, morphology, microtomography

## Abstract

**Simple Summary:**

Phoroid flies are an ancient lineage of Diptera, which includes megadiverse, widespread groups like the scuttle flies, as well as species-poor, sometimes relict, groups like flat-footed and ironic flies. The earliest fossils of phoroid flies are from Early Cretaceous. In this paper we describe a second species of the enigmatic phoroid fly genus *Lebambromyia*. The genus was erected to accommodate an extinct species, *L. acrai* Grimaldi and Cumming, from Lebanese amber deposit, dated at ca. 120 Mya. A new species, *L. sacculifera* sp. nov., is described here based on a single female specimen embedded in Myanmar “mid-Cretaceous” amber, which is over 20 Ma younger than the Lebanese outcrop, implying that this genus had a wide geographic and temporal distribution. The state of preservation of the new specimen and its study with phase contrast X-ray microtomography show that this ancient fly was characterized by a mix of ancient and modern features, such as specialized sensory areas in the antenna. Phylogenetic analyses support that *Lebambromyia* was related to flat-footed and ironic flies, but a clear phylogenetic position remains elusive.

**Abstract:**

*Lebambromyia sacculifera* sp. nov. is described from Late Cretaceous amber from Myanmar, integrating traditional observation techniques and X-ray phase contrast microtomography. *Lebambromyia sacculifera* is the second species of *Lebambromyia* after *L. acrai* Grimaldi and Cumming, described from Lebanese amber (Early Cretaceous), and the first record of this taxon from Myanmar amber, considerably extending the temporal and geographic range of this genus. The new specimen bears a previously undetected set of phylogenetically relevant characters such as a postpedicel sacculus and a prominent clypeus, which are shared with Ironomyiidae and Eumuscomorpha. Our cladistic analyses confirmed that *Lebambromyia* represented a distinct monophyletic lineage related to Platypezidae and Ironomyiidae, though its affinities are strongly influenced by the interpretation and coding of the enigmatic set of features characterizing these fossil flies.

## 1. Introduction

Diptera, or true flies, with approx. 160,000 known extant and 4000 extinct species [1] are one of the hyperdiverse groups of insects, rivalling all others in the variety of lifestyles and adaptations. During their evolutionary history, flies have exploited a wide range of niches, and colonized a wide range of habitats from the open ocean to snowy mountain peaks and even tar pits [2].

The fossil record of flies is rather extensive, since their propensity for feeding on plant exudates and their usually small size enhances their preservation in amber. Compression fossils of flies are also abundant, but their phylogenetic interpretation is sometimes more challenging due to poor quality of preservation and lack of diagnostic details. Flies are a rather ancient clade that likely originated in the Permian [3], however, the oldest unambiguous dipteran fossils are Triassic in age [4,5]. Basal Brachycera (horse flies, soldier flies, robber flies, bee flies, and relatives) differentiated much later during the Jurassic, while Eremoneura—a major lineage within Brachycera, comprising empidoids (dance flies) and Cyclorrapha (scuttle flies, flower flies, and the schizophorans)—are common in the fossil record not earlier than the Cretaceous [5,6]. Schizophora, which includes, among others, fruit flies, house flies, and blow flies, is essentially a Cenozoic radiation [3]. 

Cretaceous amber deposits preserve representatives of a disparate array of extinct lineages of uncertain phylogenetic affinity. Some lineages with low extant diversity, such as Lonchopteridae, Ironomyiidae, Platypezidae and Opetiidae have instead a rich fossil record and are well represented in Cretaceous ambers [4,7,8]. A representative of this group, *Lebambromyia acrai* Grimaldi and Cumming, 1999 was described from Lebanese amber (Lower Cretaceous, Barremian) based on two specimens and was originally assigned to Ironomyiidae, although a possible relationship with Platypezidae or a position as sister to the phoroid clade were also discussed [7]. Later, McAlpine [9] questioned the ironomyiid affinities of *Lebambromyia*, because it was not substantiated by any unambiguous apomorphy. Finally, Li and Yeates [8] recovered *Lebambromyia* as an early representative of Platypezidae in a cladistic analysis aimed at reconstructing the affinities of Ironomyiidae from Burmese amber. 

In this paper, we describe a second species of *Lebambromyia* based on an exceptionally well-preserved female specimen from the Late Cretaceous amber of Myanmar (or Burmese amber) by means of 2D/3D imaging techniques such as standard (optical) microscopes and X-ray phase contrast microtomography (XPCT), respectively. This finding, which considerably expands the temporal and paleogeographic range of *Lebambromyia*, allowed us to explicitly delimit the genus through synapomorphies and provided more insights on the phylogenetic position of this enigmatic extinct fly. Our aim here was not to reconstruct the phylogeny of phoroids but rather to test the monophyly of the hitherto monotypic genus *Lebambromyia* and to assess whether the addition of new fossils corroborates previous results. 

## 2. Materials and Methods

The specimen described herein was collected in the amber outcrops located in the Hukawng Valley of Kachin Province, ca. 100 km west of the town of Myitkyina, Myanmar, which are commonly referred as “Burmese” amber. These deposits are dated to the Late Cretaceous, 98 ± 0.6 Ma, near the Aptian/Cenomanian border, based on the radiometric dating of zircons [10], and confirmed by an ammonite trapped in an amber piece from the same locality [11].

The amber piece was first ground with emery papers of different grain sizes, and finally polished with polishing powder. Digital photomicrographs and measurements of the specimen were taken using a stereomicroscope Zeiss Stereo Discovery V 16 microscope system at the State Key Laboratory of Paleobiology and Stratigraphy, Nanjing Institute of Geology and Palaeontology, Chinese Academy of Sciences (NIGPAS). The resulting images were later digitally stacked using Helicon Focus 6.7.1 software, for a better illustration of the 3D structures.

We are aware of the ethical issues involving Burmese amber and we declare that the specimen was collected before the humanitarian crisis that started in the excavation areas in 2017 [12]. The specimen is deposited in the Nanjing Institute of Geology and Palaeontology (NIGP), Chinese Academy of Sciences at Nanjing, China, in full compliance with the recommendations of ICZN [13], and the instructions of the International Palaeoentomological Society [14]. The material examined during the present study was borrowed from the NIGP with the assurance that it had been acquired and imported in compliance with all local procedures and regulations.

Morphological terminology follows Cumming and Wood [15].

XPCT measurements were performed using the Propagation Based Imaging (PBI) setup [16]. PBI exploits self-interference of the beam propagating between the object and the detector. The sample was measured at the P05 imaging beamline operated by the Helmholtz–Zentrum Geesthacht (PETRA III storage ring, DESY) [17] using a monochromatic beam energy 30 keV. The tomography was acquired in half-acquisition mode [18] with 4001 projections and an exposure time of 0.25 s, covering a total angle range of 360 degrees. Samples were placed at 0.5 m from the imaging system [17] with pixel size of 0.64 micron. 

Data pre-processing, phase retrieval, and reconstruction were performed using the tomographic reconstruction software available at the P05 beamline [19]. Flat-/dark-fields correction was performed on raw data and each tomographic projection was normalized with the average value of the background outside the object. Binning 2 was applied over the normalized projections. The phase contrast retrieval algorithm [20] was applied to each XPCT projection. After the phase retrieval procedure, the resulting dataset was used for the 3D reconstruction of the object. The tomographic reconstruction was done with Filtered Back Projection method (FBP). The reconstructed slices exhibit an effective voxel size equal to 1.28 × 1.28 × 1.28 um^3^.

The different electron densities of the tissues were rendered as grey levels in the phase tomograms images. To independently display the different tissues, image analysis and image segmentation have been performed using the software ImageJ, Avizo Lite 9.4 and custom-made scripts implemented in ImageJ. For the 3D rendering, binning 2 was further applied over the reconstructed data leading to a final voxel size equal to 2.56 × 2.56 × 2.56 µm^3^.

The analyzed morphological dataset was originally assembled by Li and Yeates [8]. We made a few minor additions to the original matrix: besides including the new *Lebambromyia*, we added three additional binary characters, namely: character 38 (postpedicel shape), character 39 (postpedicel sacculus), and character 40 (clypeus shape) (Table 1, Table 2 and Table 3). The final version of the dataset included 41 characters, 33 of which were binary and 8 were multistate. Maximum parsimony (MP) analyses were run with TNT version 1.5 [21], implementing exhaustive tree searches, given the low number of operational taxonomic units, using the “implicit enumeration” algorithm, under both equal (EW) and implied (IW) weights. Analyses under IW were conducted with k-values ranging from 3 (default setting in TNT) to 20. Multistate characters were treated as unordered and zero-length branches were collapsed. Bremer support values were calculated under EW from 10,000 trees up to eight steps longer than the shortest as obtained from a “traditional search,” using the “trees from RAM.” Characters were optimized on the most parsimonious trees using Winclada [22].

## 3. Results

### 3.1. Systematic Palaeontology

Diptera Linnaeus, 1758

Cyclorrhapha Brauer, 1863

Phoroidea Curtis, 1833

*Lebambromyia* Grimaldi and Cumming, 1999

Diagnosis. Updated from Grimaldi and Cumming [7]. Postpedicel laterally flattened, narrowly prolonged at insertion with arista, and with prominent and rounded ventral lobe. Arista with 3 articles, of which aristomeres 1 and 2 are short. Frons with numerous small, stiff setulae and no macrosetae. Wing with Sc and R_1_ very close, particularly in middle, but not coalesced; area between Sc and R_1_ largely sclerotized; vein C extended to slightly beyond apex of M_1 + 2_; veins R_2 + 3_ and R_4 + 5_ nearly straight, not curved; dm-cu vein perpendicular (not oblique) to veins M_1 + 2_; cell cup long, more than 1.5 times length of cell bm, base of cup extended proximal well past base of cell bm and distal nearly to wing margin; no trace of vein A_2_; anal lobe very small; alula small. Female terminalia telescoping.

Apomorphies. Non-unique 7:1, 21:2; unique 38:1 (postpedicel with prominent ventral lobe).

*Lebambromyia sacculifera* Badano, Zhang, Yeates and Cerretti sp. nov.

Figure 1, Figure 2 and Figure 3

urn:lsid:zoobank.org:act:E8D1DF56-C263-427A-8C95-BE12D62BC8EC

Type material. Holotype: BA02-14122 (NIGP). Female. One specimen preserved in an amber piece cut as a parallelepiped. State of preservation: good, complete (Figure 1A,B) [Remarks: the specimen is crossed by an internal discontinuity plan hiding some portions of thorax and wing base]. 

Occurrence. Northern Myanmar, Kachin Province, Hukawng Valley, ca. 100 km west of the town of Myitkyina; Late Cretaceous (98 ± 0.6 Ma). 

Etymology. The specific epithet “*sacculifera*” is an adjective of Latin derivation, meaning “bearing a small bag”, based on the presence of a well-developed sacculus on postpedicel.

Diagnosis. Small sized phoroid fly with flattened cordate postpedicel with dorsal projection partly covering the pedicel and prominent ventral lobe; postpedicel with sacculus; mouthparts well developed, labial palp rounded; notopleuron with 3 setae.

Description. Size. Total body length 2.47 mm, wing length 2.1 mm. 

Head. Compound eye relatively large, bare (Figure 2A,B and Figure 3). Frons broad (Figure 3A). Gena relatively short in lateral view, largely occupied by compound eye (Figure 2A and Figure 3B). Fronto-genal suture deep, with a row of very short setae. Clypeus prominent, projecting forward, medially membranous (Figure 2A). Antennal pedicel conical, with a series of robust sensilla on apical margin. Postpedicel relatively large, strongly flattened, pointed at apex with a dorsal posterior projection covering the pedicel apex and with a prominent, ventral hatchet shaped lobe; lateral surface of postpedicel with a distinct, round and deep sacculus (Figure 2B). Arista bare, with 3 aristomeres, of which aristomeres 1 and 2 very short, of similar size; aristomere 3 long and thin, 2.0 times the length of pedicel + postpedicel (Figure 2B). Palpus flattened, broader apically, rounded at apex. Proboscis (when fully extended) as long as head height (Figure 2A). 

Thorax. Scutum and scutellum both strongly convex (Figure 1 and Figure 3B). Prothoracic spiracle long, elliptical. Postpronotum narrow and projecting forward. Notopleuron with 3 robust setae (Figure 1B). Dorsal surface of scutum with short, thin setae. Scutum with slightly differentiated dorsocentral setae; supralar seta absent (Figure 1B). Pleural sclerites bare. Scutellum with apical and lateral setae. Wing. Membrane with microtrichia (Figure 2C). Vein Sc very close to vein R_1_ but not fused to it; Sc and R_1_ mostly run subparallel and diverge apically (note: the wing membrane between Sc and R is folded, so that the two veins seem fused on basal two thirds, Figure 2C). Membrane between apical third of veins Sc and R_1_ slightly sclerotized, darker in color (Figure 2C). Vein Rs originating at height of crossvein h. Veins R_2 + 3_ and R_4 + 5_ almost straight. Distance between crossvein r-m and R_2 + 3_ and R_4 + 5_ fork more than 3 times length of r-m. Cell dm relatively long. Crossvein dm-cu perpendicular to M_1 + 2_. Cell cup relatively short, extending slightly beyond cell bm and ending well before wing margin (Figure 2C). Legs. Relatively thin, without tarsal modifications. 

Abdomen. Tergites 1 and 2 appear at least partly fused. Female terminalia with telescoping elements, cerci flattened, rounded at apex (Figure 2D).

Remarks. *Lebambromyia sacculifera* differs from *L. acrai* mainly in the morphology of antenna and mouthparts. The new species is characterized by a broad postpedicel provided with a dorsal projection, covering the apex of pedicel, and a relatively large ventral lobe (Figure 2A,B). On the contrary, *L. acrai* is provided with a cordate postpedicel, without a dorsal projection and with a comparatively smaller ventral lobe. *Lebambromyia sacculifera* is characterized by a prominent lower facial margin and well-developed mouthparts, with rounded labial palp (Figure 2A). In contrast, *Lebambromyia acrai* has a short, not projecting, lower facial margin and vestigial mouthparts, but it is still provided with a normally developed labial palp, which is apically pointed [7]. The notopleuron of *Lebambromyia sacculifera* bears 3 setae (4 in *L. acrai*) (Figure 1B).

### 3.2. Phase Contrast XPCT

XPCT images overcome the disruption crossing the amber piece in proximity of the inclusion, which distorts the specimen when it is observed under optical instruments, hiding relevant key features. XPCT image reveals that the head is perfectly preserved, showing a relatively broad frons, a prominent clypeus and dichoptic, well-spaced, compound eyes (Figure 3A; Appendix A). The frontoclypeal suture relatively deep, creating a strong depression around the insertion of mouthparts (Figure 3A). The cervical membrane, which articulates the head and thorax, and the tentorium are rendered in some detail in Figure 3B. The strongly convex scutellum and the legs bear traces of internal structures, which are likely muscle remains (Figure 3B). The XPCT reconstructions (Figure 3B) suggest that the abdominal tergites 1 and 2 were laterally fused, a condition only shared with Platypezidae, Ironomyiidae, and the Eumuscomorpha [9]. However, the state of preservation might influence the detection of the latter character.

### 3.3. Phylogenetic Analysis

The MP analyses yielded one most parsimonious tree (Figure 4) under the whole range of weighting schemes employed (EW: tree length: 71, Consistency index: 0.704, Retention index: 0.809). *Lebambromyia* was recovered as part of a clade also including Platypezidae and Ironomyiidae. The monophyly of this clade relied on three unique apomorphies (8:1, supra-alar setae absent; 26:1, cell cua longer than 1.3 times of cell bm; 29:2, C extending to M_1_); under equal weights this clade obtained a Bremer support value of 2. Platypezidae were monophyletic based on two unique apomorphies (34:1; 36:1) and were recovered as sister to *Lebambromyia* + Ironomyiidae. The clade including *Lebambromyia* and Ironomyiidae was in turn supported by one unique (39:1, sacculus present) and one non unique (10:1) apomorphies. *Lebambromyia* was retrieved as monophyletic based on one unique (38:1, postpedicel with a ventral flattened lobe) and two (7:1, 21:2) non unique apomorphies. The monophyly of Ironomyiidae relied on two unique apomorphies (14:1, Sc in apical half of wing; 15:1, Sc and R_1_ fused medially and long). Under equal weights monophyly of both, *Lebambromyia* and Ironomyiidae obtained a Bremer support value of 2 (Figure 4).

## 4. Discussion

The affinities of flies embedded in Mesozoic ambers are often enigmatic due the quality of preservation, the absence of appreciable apomorphies, or the presence of combinations of morphological characters absent in living taxa. These fossil flies are often stem-groups of successful radiations or are representatives of lineages that are now relict, both in terms of diversity and geographic distribution [7]. Phoroidea *sensu* Pape et al. [1] (= Platypezoidea *sensu* Wiegmann et al. [3]; Brown et al., [23]; Amorim et al. [24]) are significant in this respect, because they have a rather extensive Mesozoic fossil record and they include both ancient groups characterized by low extant diversity, often with a relict distribution, e.g., Ironomyiidae (3 species, limited to Australia), Opetiidae (4 species, limited to the Palaearctic and Chile) and Lonchopteridae (50 species, worldwide), and highly successful, hyperdiverse, widespread clades such as Phoridae (over 6000 described species, worldwide) [2,4,9,24,25,26,27]. Between these extremes, Phoroidea also include Platypezidae, a medium-sized and widespread group including 250 known species [3,28].

Recent phylogenetic analyses of the Phoroidea based on both molecular and morphological data and a diverse array of ingroup taxa have yielded diverging topologies, e.g., [3,7,8,27,28]. Notably, Wiegmann et al. [3] reconstructed the internal relationships among the phoroid families as a successive splitting of lonchopterids, opetiids, platypezids, ironomyiids and phorids; whereas Tkoč et al. [28] reconstructed phorids sister to a grade of platypezids, from which opetiids arose. More recently, the morphology-based cladistic analysis by Li and Yeates [8], which also included a selection of fossil taxa, retrieved ironomyiids sister to the remaining phoroids and platypezids sister to a clade composed of opetiids, lonchopterids and phorids. Li and Yeates [8] investigated for the first time the phylogenetic affinities of *Lebambromyia acrai* using an explicit, cladistic approach and recovered it as an early platypezid, although with weak branch supports. In fact, our analysis, although based on nearly the same data set by Li and Yeates [8], retrieved Phoroidea split in two clades: (platypezids (*Lebambromyia*, ironomyiids)) sister to (opetiids (lonchopterids, phorids)), again demonstrating that fossil taxa are often critical to reconstruct the evolutionary history of life. 

The phoroid affinities of *Lebambromyia* were recognized since its original description, although its actual relationships remained somewhat enigmatic. Grimaldi and Cumming [7] suggested a close relationship between *Lebambromyia* and Ironomyiidae mostly based on the presence of a “pterostigma,” i.e., a sclerotization of the wing membrane between Sc and R_1_. However, as pointed out by McAlpine [9], a similar, lightly sclerotized area is also present in several Platypezidae, while *Lebambromyia* lacked the apomorphies of Ironomyiidae, i.e., the fusion of veins Sc and R_1_ and shape of pedicel and postpedicel. 

The discovery of a perfectly preserved specimen of an undescribed *Lebambromyia* from Myanmar amber showed that this fly was characterized by a deep postpedicel sacculus (character 39, state 1) which is a character state shared with Ironomyiidae and Eumuscomorpha (i.e., syrphoid grade + Schizophora). The sacculus is a deep cuticular invagination on the external side of the postpedicel, housing chambers filled with sensilla, likely increasing the chemoreceptive area, thus likely overcoming the reduction of the antenna characteristic of most Brachycera [29,30]. Despite the number and shape of sacculi vary across these three groups, i.e., one in *L. sacculifera*, two in *Ironomyia* (unknown in extinct ironomyiids), one or more in eumuscomorphs, McAlpine [9,29] treated it as a homologous feature. However, the character state optimization on our tree reconstructed the presence of postpedicel sacculus as a synapomorphy supporting monophyly of *Lebambromyia* + Ironomyiidae (Figure 4). If the latter reconstruction is true, the ancestor of Eumuscomorpha (i.e., the sister clade of Phoroidea) likely evolved postpedicel sacculi independently. The inclusion of *Lebambromyia sacculifera* in the morphological dataset “reshuffled the deck” on our understanding of the relationships of this taxon, casting doubts on its Platypezidae affiliation, as suggested by Li and Yeates [8], though not supporting an affinity with Ironomyiidae without any reasonable doubt. Interestingly, our results support the view of McAlpine [9], who argued that the placement of this enigmatic fly into a distinct family-ranked group was likely needed. 

“Burmese” amber preserves one of the most diverse and most extensively studied Cretaceous biota [31]. During the deposition of amber outcrops, the Burma Terrane was a Trans-Tethyan island arc at equatorial latitudes, suggesting that the faunal assemblage might have included insular endemics [32]. However, a comparison between the amber deposits of the Lower Cretaceous (Lebanese amber) and of the Late Cretaceous (e.g., Spanish, French and Myanmar ambers) shows that the differences in faunal composition among these outcrops were mostly age dependent [33]. Lebanese amber significantly differs from younger deposits, while French and Myanmar ambers, which are of similar age, also share several common elements [33]. Nevertheless, Lebanese and Myanmar ambers still have a few taxa in common despite the chronological and paleogeographic differences, such as the beetle-like cockroach *Cratovitisma* Bechly, 2007, the mantis *Burmantis* Grimaldi, 2003 and the phylogenetically enigmatic fly family Chimeromyiidae [6,34,35]. 

The discovery of *Lebambromyia* in Burmese amber is significant from a biogeographic point of view, representing a new shared faunal element between these deposits and suggesting that further common elements might await discovery. The new finding also considerably expands the geographic and temporal range of *Lebambromyia*, implying that this genus occurred for more than 25 Ma, and reached a wide geographic distribution also colonizing islands. On the other hand, *Lebambromyia* appears morphologically conservative despite its extensive temporal range, as *L. acrai* and *L. sacculifera* differ mainly in details of antenna and mouthparts. 

## 5. Conclusions

Cretaceous ambers from the Barremian to the Cenomanian document a diversification phase for eremoneuran flies, a glimpse of the massive radiation of this group yet to come. Phoroid lineages are particularly well represented with stem-groups or early representatives of nowadays relict clades. The presence of *Lebambromyia* in both Lebanese and Myanmar ambers, suggest these ancient flies were not short-living evolutionary “experiments” but successful groups in themselves, sometimes lasting for millions of years.

## Figures and Tables

**Figure 1 insects-12-00354-f001:**
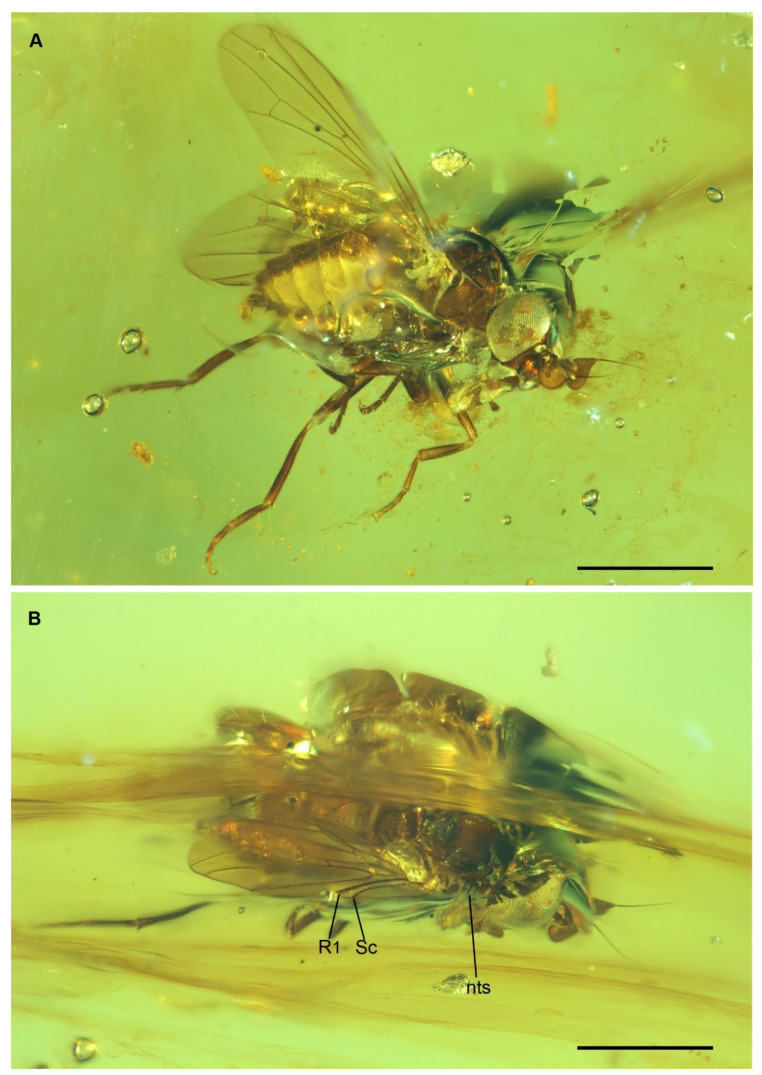
*Lebambromyia sacculifera* sp. nov., holotype, BA02-14122, habitus. (**A**) lateral view; (**B**) dorsal view. Abbreviations: Sc, Subcosta; R1, Radius; nts, notopleural setae. Scale bar: 1 mm.

**Figure 2 insects-12-00354-f002:**
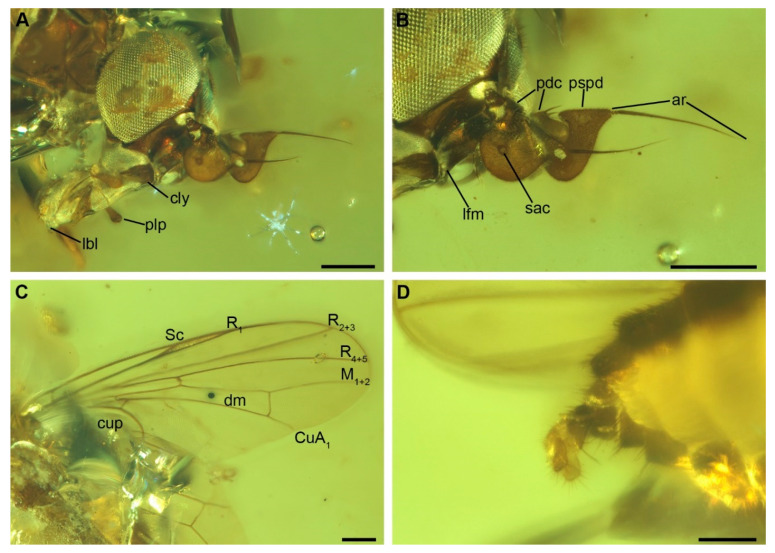
*Lebambromyia sacculifera* sp. nov., holotype, BA02-14122. (**A**) detail of the head and mouthparts, lateral view; (**B**) detail of the front and antennae; (**C**) wing; (**D**) terminalia. Abbreviations: ar, arista; cly, clypeus, lbl, labellum; lfm, lower facial margin; pdc, pedicel; plp, palpus; pspd, postpedicel; sac, sacculus; Sc, Subcosta; R, Radius (and its branches); M, Media; CuA1, Cubitus anterior; dm, discal-medial cell; cup, anal cell. Scale bar: 200 µm.

**Figure 3 insects-12-00354-f003:**
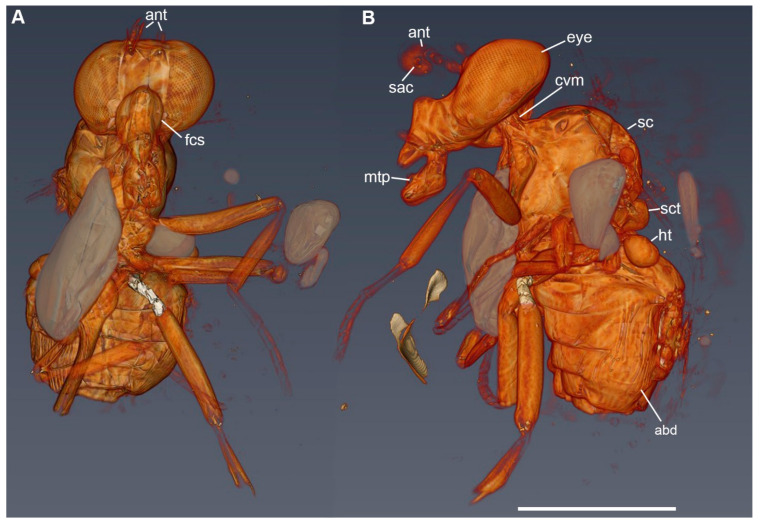
*Lebambromyia sacculifera* sp. nov., holotype, BA02-14122, 3D rendering of XPCT image. Embedded air bubbles in grey. (**A**) ventral view; (**B**) lateral view. Abbreviations: abd, abdomen; ant, antenna; cvm, cervical membrane; eye, compound eye; fcs, frontoclypeal suture; mtp, mouthparts; ht, haltere; sac, sacculus; sc, scutum; sct, scutellum. Scale bar: 1 mm.

**Figure 4 insects-12-00354-f004:**
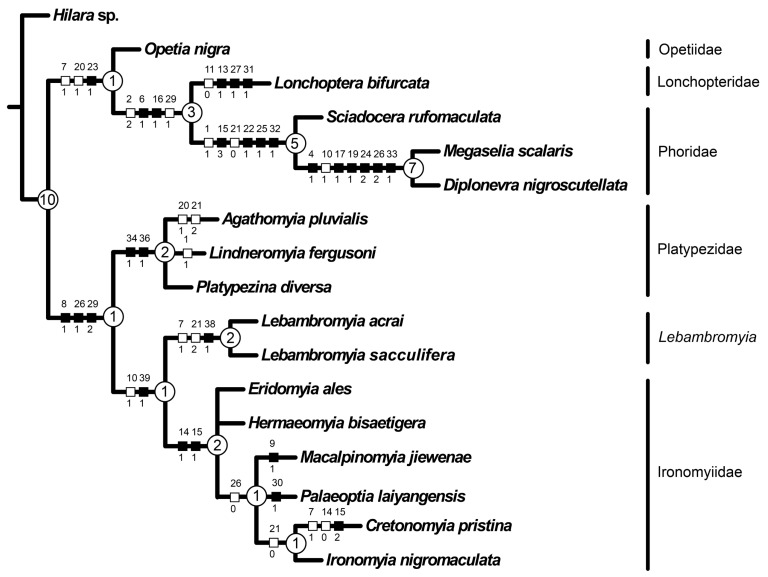
Most parsimonious tree obtained under equal weight. Inferred character state changes are mapped over branches, white squares indicate non-unique apomorphies while black squares are unique apomorphies. Numbers at nodes are Bremer support values.

**Table 1 insects-12-00354-t001:** Characters added to the morphological dataset of Li and Yeates [8] for the phylogenetic analysis.

38.	Postpedicel, prominent ventral lobe: (0) absent; (1) present.
39.	Postpedicel, sacculus: (0) absent; (1) present.
40.	Clypeus: (0) not prominent; (1) flattened against head ventral surface; (2) prominent, projecting forward.

**Table 2 insects-12-00354-t002:** Scoring of the additional characters in the morphological dataset of Li and Yeates [8].

Taxon	Characters		
	38	39	40
Empididae (outgroup)			
*Hilara* sp.	0	0	0
PHOROIDEA			
Ironomyiidae			
*Cretonomyia pristina* McAlpine, 1973	0	?	?
*Eridomyia ales* Mostovsky, 1995	?	?	?
*Hermaeomyia bisaetigera* Mostovsky, 1995	?	?	?
*Ironomyia nigromaculata* White, 1916	0	1	2
*Macalpinomyia jiewenae* Li and Yeates, 2019	?	?	?
*Palaeoptia laiyangensis* Zhang, 1987	?	?	?
Lonchopteridae			
*Lonchoptera bifurcata* Fallén, 1810	0	0	0
Opetiidae			
*Opetia nigra* Meigen, 1830	0	0	0
Phoridae			
*Diplonevra nigroscutellata* (Malloch, 1925)	0	0	0
*Megaselia scalaris* (Loew, 1866)	0	0	0
*Sciadocera rufomaculata* White, 1916	0	0	0
Platypezidae			
*Agathomyia pluvialis* Chandler, 1994	0	0	1
*Lindneromyia fergusoni* (Tonnoir, 1925)	0	0	1
*Platypezina diversa* (Johnson, 1923)	0	0	1
Unassigned to family			
*Lebambromyia acrai* Grimaldi and Cumming, 1999	1	?	?
*Lebambromyia sacculifera* sp. nov.	1	1	2

**Table 3 insects-12-00354-t003:** Scoring of *Lebambromyia sacculifera* in the morphological dataset of Li and Yeates [8].

Taxon	Characters																				
*Lebambromyia* *sacculifera*	0	1	2	3	4	5	6	7	8	9	10	11	12	13	14	15	16	17	18	19	20
	?	0	2	1	0	?	?	1	1	?	1	?	0	2	0	0	0	0	1	0	0
	21	22	23	24	25	26	27	28	29	30	31	32	33	34	35	36	37	38	39	40	
	2	0	0	1	0	1	?	0	2	0	0	?	0	0	1	0	?	1	1	2	

## Data Availability

Data is contained within the article or Appendix A.

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
