# Peer review of "Discovery of Lebambromyia in Myanmar Cretaceous Amber: Phylogenetic and Biogeographic Implications (Insecta, Diptera, Phoroidea)"

_insects, 2021, doi:10.3390/insects12040354_

Round 1

Reviewer 1 Report

Basic reporting

It is an interesting and well written manuscript, meeting the high standards of the journal Insects. The authors follow standard procedures of systematic paleoentomology. The fossil specimen is sufficiently described and preserved in a public collection. The figures are of high quality, with appropriate legends. Using of X-ray microtomography gives additional attractiveness to the paper. The phylogenetic analysis presented, however, shows some problematic issues and should be improved. The new species appears as valid but its practical identification in the future may be difficult since the description is based on a single female.

Specific issues

Personally I do not consider a good taxonomic practice to describe a new species based just on a single female specimen. This may potentially bring more problems than benefits. In the groups like Diptera, male terminalia usually provide the most important species-specific characters, as well as characters relevant to phylogenetic analyses. Actually, a probability of future correct association of a named fossil female with a male of the same species is very low. The authors should specify why they decided to follow such a practice and how their new species can be recognized in the future, when, e.g., congeneric males (of two or more species differing just in male terminalia) are found in Burmese amber. Their opinion on this issue should be included in the manuscript.

The family assignment of the genus has not been resolved but possibly something more specific than Cyclorrhapha could be included in the title (e.g. Phoroidea or Platypezoidea).

The wing venation of Lebambromyia sacculifera appears rather different from Lebambromyia acrai (e.g. position of r-m vs. point of furcation of R2+3 and R4+5, the shape of cells dm and cup), so that one could consider them as separate genera. Such a possibility is not discussed in the manuscript and these differences are not mentioned under Diagnosis (or Remarks) of the new species.

Phylogenetic analysis: I strongly recommend including the entire data matrix (character scoring) in the present manuscript, to avoid the necessity to seek after previous papers. The same applies to the description and explanation of particular characters.

There are several problematic issues in the phylogenetic analysis presented. They should be corrected and the analysis re-ran. Here are some examples:

Character 5: There are strong fronto-orbitals in the females of both Agathomyia and Platypezina, but not in their holoptic males. Apparently, a female of Agathomyia and a male of Platypezina have been studied. This evidently led Li and Yeates (2019) to placing Platypezina in a clade with Lindneromyia instead of with Agathomyia. Such errors in the dataset should be corrected.

Character 20: The interpretation of pterostigma is somewhat unclear. It apparently refers only to a darkening of the subcostal cell as in Lebambromyia. Li & Yeates listed this for Agathomyia but not the other two platypezid genera included, but Platypezina has a well-developed true pterostigma extending beyond R1 as in Microsania, see supplementary figures in Tkoc et al. (2017), and this darkening is also apparent in some Lindneromyia spp. and other Platypezinae. This character possibly deserves a more detailed coding than just presence or absence.

Character 29: In the analysis by Li and Yeates (2019), only Lebambromyia + three genera of Platypezidae are coded as 2. On the contrary, in the present manuscript, six other genera (the entire family Ironomyiidae) are apparently coded as 2 (see Fig. 4). This should be corrected or explained.

Character 34 is included on the stem of the Platypezidae in the present tree, but it only applies to Lindneromyia of the three included genera, as the representative of Platypezinae. The same error is already in Li & Yeates (2019).

Characters 38-40 are not coded for Opetia nigra, although it is an extant species, present in various collections. The authors should contact specialists on Platypezidae and ask them to provide relevant character states. It would be also interesting to know where Microsania (a related extant taxon) would fit in the trees.

Other suggestions:

Table 2: complete taxonomic names should be provided (with authors and dates), also some other details could be added (males or females studied, number of specimens, variation)

Lines 75 and 76 should be connected

Line 300: not [3,28] but [3,27]

Author Response

Reply to reviewer 1

It is an interesting and well written manuscript, meeting the high standards of the journal Insects. The authors follow standard procedures of systematic paleoentomology. The fossil specimen is sufficiently described and preserved in a public collection. The figures are of high quality, with appropriate legends. Using of X-ray microtomography gives additional attractiveness to the paper. The phylogenetic analysis presented, however, shows some problematic issues and should be improved. The new species appears as valid but its practical identification in the future may be difficult since the description is based on a single female.

  • We thank the reviewer for the useful comments.

Specific issues

Personally I do not consider a good taxonomic practice to describe a new species based just on a single female specimen. This may potentially bring more problems than benefits. In the groups like Diptera, male terminalia usually provide the most important species-specific characters, as well as characters relevant to phylogenetic analyses. Actually, a probability of future correct association of a named fossil female with a male of the same species is very low. The authors should specify why they decided to follow such a practice and how their new species can be recognized in the future, when, e.g., congeneric males (of two or more species differing just in male terminalia) are found in Burmese amber. Their opinion on this issue should be included in the manuscript.

  • We agree with the reviewer that describing a new species based on male specimens is (often) preferable, though not always. There are groups where females are taxonomically more informative than males (this is common for Hymenoptera, but also many Diptera). We decided to name a new Lebambromyia not just to put a label on a long dead fly, but because this specimen was a representative of a key, extinct and puzzling lineage. Lebambromyia was erected to allocate a new species based on female specimens encased in a piece of Lebanese amber. So, the genus is known only on female specimens which are readily comparable; an issue would raise once (and if) a male will ever be discovered. The suggestion of “waiting for more material” is negligible (who decides how many specimens is enough) especially for fossil material because every single piece would be critical to understand significant parts of the story of life on Earth. This new species shows a whole new set of features that will allow future comparisons. Moreover, diagnostic features of the genitalia are very rarely appreciable in specimen preserved in Burmese amber, even applying the most sophisticated imaging techniques. In the end, Lebambromyia sacculifera is characterized by a set of characters that is unique among known living and extinct Diptera. This is the fundamental requirement for erecting a new species.

The family assignment of the genus has not been resolved but possibly something more specific than Cyclorrhapha could be included in the title (e.g. Phoroidea or Platypezoidea).

  • We thank the reviewer for this suggestion, and we modified the title accordingly.

The wing venation of Lebambromyia sacculifera appears rather different from Lebambromyia acrai (e.g. position of r-m vs. point of furcation of R2+3 and R4+5, the shape of cells dm and cup), so that one could consider them as separate genera. Such a possibility is not discussed in the manuscript and these differences are not mentioned under Diagnosis (or Remarks) of the new species.

  • The reviewer probably refers here to Fig. 4C. However, the position of the specimen within the amber piece, the refraction properties of amber and the internal impurities make difficult to fully appreciate the wing shape, also distorting the vein origin. The specimen is inclined along the vertical axis (so the wing is seen obliquely) so we gave images of the specimen from all possible angles to avoid this problem. For example, if one check Fig. 1A, it will notice that cell dm has the same shape of that acrai. The wing characters mentioned above are all very similar to L. acrai—though the shape of cell cup is not appreciable due to impurities near the body—or differ in a such minor ways that do not support the erection of a new genus.

Phylogenetic analysis: I strongly recommend including the entire data matrix (character scoring) in the present manuscript, to avoid the necessity to seek after previous papers. The same applies to the description and explanation of particular characters.

  • The data matrix was already published by Li and Yeates (2019), like the list of characters, descriptions, characters figures and examined species. We refer to that work, and we cannot re-publish it.

There are several problematic issues in the phylogenetic analysis presented. They should be corrected and the analysis re-ran. Here are some examples:

  • We thank the reviewer for noticing these issues. We corrected the matrix whenever possible, and we re-ran the analysis. However, the resulting topology and supports are not significantly changed as the proposed corrections only refer to the internal resolution of a terminal including 3 taxa (i.e., Platypezidae), thus they do not influence the phylogenetic backbone, nor the position of Lebambromyia. The aim of the analysis is to test the monophyly and the affinities of Lebambromyia and not to reconstruct the phylogeny of other groups such as Platypezidae or Opetiidae, this is well beyond the aim of the present work and will require a dedicated taxon sampling and set of characters.

Character 5: There are strong fronto-orbitals in the females of both Agathomyia and Platypezina, but not in their holoptic males. Apparently, a female of Agathomyia and a male of Platypezina have been studied. This evidently led Li and Yeates (2019) to placing Platypezina in a clade with Lindneromyia instead of with Agathomyia. Such errors in the dataset should be corrected.

  • We thank the reviewer for noticing this error. We corrected the character state of Platypezina, though this is not influential to the rest of the analysis.

Character 20: The interpretation of pterostigma is somewhat unclear. It apparently refers only to a darkening of the subcostal cell as in Lebambromyia. Li & Yeates listed this for Agathomyia but not the other two platypezid genera included, but Platypezina has a well-developed true pterostigma extending beyond R1 as in Microsania, see supplementary figures in Tkoc et al. (2017), and this darkening is also apparent in some Lindneromyia spp. and other Platypezinae. This character possibly deserves a more detailed coding than just presence or absence.

  • The interpretation of pterostigma follows Li and Yeates (2019). Please, note than in the dataset the presence is marked with “0” and absence with “1” (derived character state). So, the character is coded as present in both Platypezina and Lindneromyia ( fergusoni is one of the representatives of the genus with pterostigma), and as absent in Agathomyia. See Supporting information S3 of Li and Yeates (2019).

Character 29: In the analysis by Li and Yeates (2019), only Lebambromyia + three genera of Platypezidae are coded as 2. On the contrary, in the present manuscript, six other genera (the entire family Ironomyiidae) are apparently coded as 2 (see Fig. 4). This should be corrected or explained.

  • No change has been made to the original dataset of Li and Yeates (2019), except for the addition of sacculifera and of characters 38, 39, 40. Character 29 (Costa) cannot be distinguished in fossil Ironomyiidae and is coded as missing “?” also in the present version of the dataset. Character state 29:2 is recovered as a unique synapomorphy of Platypezidae + (Lebambromyia + Ironomyiidae) based on the optimization made by the maximum parsimony algorithm implemented in TNT.

Character 34 is included on the stem of the Platypezidae in the present tree, but it only applies to Lindneromyia of the three included genera, as the representative of Platypezinae. The same error is already in Li & Yeates (2019).

  • Character 34 (hind tarsus) is flattened in Agathomyia (Chandler & Shatalkin in Manual of Palaearctic Diptera, 1998), Lindneromyia (Li and Yeates, 2019, supporting information S3) and Platypezina (Johnson, 1923; Kessel, 1961). We are not referring here to the specialized female tarsus with soles, which is a synapomorphy of Platypezinae. In any case, this is again an apomorphic character referring to one clade, whose internal resolution is beyond the aim of the study and will not influence the backbone or the affinities of Lebambromyia.

Characters 38-40 are not coded for Opetia nigra, although it is an extant species, present in various collections. The authors should contact specialists on Platypezidae and ask them to provide relevant character states. It would be also interesting to know where Microsania (a related extant taxon) would fit in the trees.

  • We thank the reviewer for noticing this error, which involves Table 2 and not the real phylogenetic dataset, where Opetia was coded for all characters (so no influence on the topology). We have corrected the table. The relationships of Microsania are beyond the aim of the present article, which is focused on Lebambromyia.

Other suggestions:

Table 2: complete taxonomic names should be provided (with authors and dates), also some other details could be added (males or females studied, number of specimens, variation)

  • Authors’ names have been added. For further data see Li and Yeates (2019).

Lines 75 and 76 should be connected.

  • Accepted.

Line 300: not [3,28] but [3,27]

  • Accepted.

Reviewer 2 Report

The manuscript describes the second species of a very interesting phorid genus, Lebambromyia.

The manuscript is original and has scientific relevance.

The introduction is clear, the methodology and analysis are adequate and the results and discussion are relevant.

Overall, it has excellent quality and great scientific interest.

The illustrative material is of excellent quality.

I believe that the manuscript can be published in the form in which it is presented.

Author Response

We thank the reviewer for the comments and appreciation of the manuscript.